# Notoginsenoside R1 Attenuates Cisplatin-Induced Ototoxicity by Inducing Heme Oxygenase-1 Expression and Suppressing Oxidative Stress

**DOI:** 10.3390/ijms252111444

**Published:** 2024-10-24

**Authors:** Yi-Chun Lin, Yi-Jung Ho, Yuan-Yung Lin, Ai-Ho Liao, Chao-Yin Kuo, Hang-Kang Chen, Hsin-Chien Chen, Chih-Hung Wang, Cheng-Ping Shih

**Affiliations:** 1Department of Otolaryngology-Head and Neck Surgery, Tri-Service General Hospital, National Defense Medical Center, Taipei 11490, Taiwan; lyc_1023@yahoo.com.tw (Y.-C.L.); yking1109@gmail.com (Y.-Y.L.); chefsketchup@hotmail.com (C.-Y.K.); hwalongchen@yahoo.com.tw (H.-K.C.); acolufreia@yahoo.com.tw (H.-C.C.); chw@ms3.hinet.net (C.-H.W.); 2School of Pharmacy, National Defense Medical Center, Taipei 11490, Taiwan; ejung330@gmail.com; 3Graduate Institute of Life Science, National Defense Medical Center, Taipei 11490, Taiwan; 4Graduate Institute of Biomedical Engineering, National Taiwan University of Science and Technology, Taipei 10607, Taiwan; aiho@mail.ntust.edu.tw; 5Department of Otolaryngology, Taipei City Hospital, Taipei 103212, Taiwan

**Keywords:** cisplatin, cochlea, hair cells, notoginsenoside, ototoxicity, oxidative stress

## Abstract

Cisplatin-induced ototoxicity occurs in approximately half of patients treated with cisplatin, and pediatric patients are more likely to be affected than adults. The oxidative stress elicited by cisplatin is a key contributor to the pathogenesis of ototoxicity. Notoginsenoside R1 (NGR1), the main bioactive compound of *Panax notoginseng* saponins, has antioxidant and antiapoptotic effects. This study investigated the ability of NGR1 to protect against cisplatin-induced damage in auditory HEI-OC1 cells and neonatal murine cochlear explants. The viability of HEI-OC1 cells treated with NGR1 and cisplatin was greater than that of cells treated with cisplatin alone. The results of Western blots and immunostaining for cleaved caspase-3 revealed that the level of cleaved caspase-3 in the cells treated with cisplatin was repressed by NGR1. NGR1 attenuated cisplatin-induced cytotoxicity in HEI-OC1 cells. Intracellular reactive oxygen species (ROS) were detected with a DCFDA assay and immunostaining for 4-HNE. The result revealed that its expression was induced by cisplatin and was significantly reduced by NGR1. Moreover, NGR1 can promote heme oxygenase-1 (HO-1) expression at both the mRNA and protein levels. ZNPPIX, an HO-1 inhibitor, was administered to cisplatin-treated cells to investigate the role of HO-1 in the protective effect of NGR1. The suppression of HO-1 activity by ZNPPIX markedly abolished the protective effect of NGR1 on cisplatin-treated cells. Therefore, NGR1 protects cells from cisplatin-induced damage by activating HO-1 and its antioxidative activity. In cochlear explants, NGR1 protects cochlear hair cells and attenuates cisplatin-induced ototoxicity by inhibiting ROS generation. In the group treated with cisplatin alone, prominent loss of outer hair cells and severe damage to the structure of the stereociliary bundles of inner and outer hair cells were observed. Compared with the group treated with cisplatin alone, less loss of outer hair cells (*p* = 0.009) and better preservation of the stereociliary bundles of hair cells were observed in the group treated with cisplatin and NGR1. In conclusion, these findings indicate that NGR1 can protect against cisplatin-induced ototoxicity by inducing HO-1 expression and suppressing oxidative stress.

## 1. Introduction

Cisplatin is the most commonly used platinum-based chemotherapeutic drug for treating numerous malignancies. Ototoxicity is a serious side effect of cisplatin and manifests as irreversible hearing loss and tinnitus. The incidence of cisplatin-induced ototoxicity ranges from 36% to 60%, and pediatric patients are more prevalently affected than adults [1,2,3,4]. Cisplatin accumulation within the cochlea leads to varying degrees of injury to the organ of Corti, spiral ganglion neurons and stria vascularis [4]. High-frequency sensorineural hearing loss occurs at an early stage because the outer hair cells (OHCs) of the basal turn of the cochlea are more susceptible to the toxicity of cisplatin [2,5]. Cisplatin primarily passes from the blood through the vasculature of the stria vascularis via transporters in marginal cells into the scala media of the cochlea. Subsequently, cisplatin diffuses across the apical membrane or is transported via various cation transport proteins and mechano-electrical transduction channels into cochlear hair cells. It induces DNA and mitochondrial damage and activates cellular apoptosis [4]. The lesions of hair cells depend on the accumulated dosage of administered cisplatin and include disarray of the stereocilia of hair cells, loss of stereociliary bundles and OHC loss [2,5]. Several mechanisms contribute to cisplatin-induced ototoxicity [2,4]. Oxidative stress in the cochlea elicited by cisplatin is a key contributor to the pathogenesis of ototoxicity [2,4,6]. Cisplatin induces the generation of ROS (reactive oxygen species) via the activation of NADPH oxidase 3, xanthine oxidase and calcium influx in the cochlea [2,4]. Damaged DNA and mitochondrial dysfunction contribute to ROS formation. The activity of antioxidant enzymes is inhibited by cisplatin [4]. Furthermore, inducible nitric oxide synthase (iNOS) activated by cisplatin results in the production of reactive nitrogen species (RNS) and nitrative stress in the cochlea [2]. Heme oxygenase-1 (HO-1) has antioxidant activity and contributes to redox homeostasis in hair cells [7,8]. The induction of HO-1 expression has a protective effect on cisplatin-induced damage to the organ of Corti [9]. These findings suggest that therapeutic agents that activate HO-1 signaling could have the potential to prevent cisplatin-induced ototoxicity. The antioxidant sodium thiosulfate is approved for clinical application to treat cisplatin-induced hearing loss in pediatric patients [4]. However, approximately 30% of patients receiving sodium thiosulfate still suffer from hearing loss [10,11]. Other effective agents need to be discovered for better prevention outcomes.

*Panax notoginseng* saponins (PNSs) are extracted from the herbal material *Panax notoginseng* [12]. PNSs have several biological functions, including anti-inflammatory effects, anticancer effects, antineurotoxic effects and inhibitory effects on ischemic injuries [13,14]. In animal studies, PNS attenuates cisplatin-induced nephrotoxicity by inhibiting apoptosis and oxidative stress [15,16]. Fei et al. observed the protective effect of PNS on cisplatin-induced cytotoxicity in HEI-OC1 auditory cells [17]. Pretreatment with PNS reduces intracellular ROS levels and activates AKT/nuclear factor erythroid 2-related factor 2 (Nrf2) signaling and HO-1 expression in cells. Notoginsenoside R1 (NGR1) is the main bioactive compound of PNSs and exerts therapeutic and protective effects on various organ systems [12,18]. NGR1 attenuates ischemia-reperfusion injury mainly through the inhibition of oxidative stress, apoptosis and inflammation [19]. The antioxidant activity of NGR1 involves the inhibition of the oxidative stress pathway and nicotinamide adenosine dinucleotide phosphate oxidase, the preservation of the mitochondrial membrane potential and the activation of antioxidant enzymes [18,19]. NGR1 upregulates the Nrf2/HO-1 signaling pathway to protect against diabetic nephropathy, cardiomyopathy and osteoarthritis [20,21,22]. Nrf2/HO-1 signaling plays an important role in maintaining redox homeostasis and has a protective effect on cisplatin-induced ototoxicity [23,24]. However, whether NGR1 can ameliorate cisplatin-induced ototoxicity is unknown. In this study, we first demonstrated that NGR1 attenuates cisplatin-induced damage to, and oxidative stress in, auditory cells and neonatal murine cochlear explants.

## 2. Results

### 2.1. NGR1 Inhibits Cisplatin-Induced Apoptosis in Auditory HEI-OC1 Cells

HEI-OC1 cells were treated with various concentrations of NGR1 for 24 or 48 h to evaluate the impact of NGR1 on cell viability (Figure 1A,B). Compared with untreated cells, cells treated with 50 μM NGR1 presented lower viability (*p* < 0.005). The administration of NGR1 at concentrations ranging from 10 to 40 μM did not lead to cytotoxicity in HEI-OC1 cells. First, the effect of 30 μM NGR1 on cell viability in cisplatin-treated HEI-OC1 cells was investigated (Appendix A). There was no significant difference in cell viability between the group with cisplatin alone and the group with cisplatin and 30 μM NGR1 (*p* = 0.8217). Therefore, the concentration of NGR1 used in subsequent experiments was 40 μM. The protective effect of 40 μM NGR1 on cisplatin-induced cytotoxicity was investigated (Figure 1C). The viability of the cells in the groups treated with cisplatin alone or with NGR1 and cisplatin was compared. A significant difference was observed between the two groups (78.4 ± 1.12% vs. 88.4 ± 1.65%, *p* < 0.001). The administration of NGR1 rescued the viability of HEI-OC1 cells treated with cisplatin. These findings suggested that NGR1 protected HEI-OC1 cells from cisplatin-induced cytotoxicity. The effects of cisplatin and NGR1 on the mitochondrial mass in HEI-OC1 cells were evaluated (Figure 2). Compared with the control, less staining of mitochondria was detected in the cells after cisplatin treatment. Pretreatment with NGR1 led to the restoration of the mitochondrial mass that was reduced by cisplatin. These results revealed that NGR1 may protect HEI-OC1 cells from the cisplatin-induced reduction in the mitochondrial mass. The protective effect of NGR1 on cisplatin-induced apoptosis in HEI-OC1 cells was subsequently investigated (Figure 3). The TUNEL assay revealed more TUNEL-positive HEI-OC1 cells after cisplatin treatment than after cisplatin and NGR1 treatment. Furthermore, the results of Western blotting and immunostaining for cleaved caspase-3 revealed that the level of cleaved caspase-3 induced by cisplatin was attenuated by NGR1 in cells. Taken together, these findings indicate that NGR1 inhibits cisplatin-induced apoptosis in auditory cells.

### 2.2. NGR1 Alleviates Cisplatin-Induced Cytotoxicity in Auditory Cells by Ameliorating Oxidative Stress and Inducing HO-1 Expression

Oxidative stress is the main contributor to the pathogenesis of cisplatin-induced ototoxicity [2]. NGR1 can suppress oxidative stress during ischemia/reperfusion injury in several organs [18,19]. Therefore, the antioxidant effect of NGR1 on cisplatin-induced ROS generation in HEI-OC1 cells was investigated (Figure 4). Intracellular ROS levels were detected with a DCFDA assay, and immunostaining for 4-HNE was increased following cisplatin treatment. NGR1 significantly reduced ROS levels in cisplatin-treated cells (*p* < 0.001). HO-1 exerts an antioxidant response to cisplatin-induced oxidative damage and can be activated to protect hearing against cisplatin-induced ototoxicity in animals [8,9,24]. Previous studies have shown that PNS upregulates HO-1 expression in auditory cells and alleviates cisplatin-induced ROS generation [17]. We evaluated HO-1 expression in HEI-OC1 cells treated with NGR1 and/or cisplatin (Figure 5A,B). According to the Western blot analysis of the HO-1 protein, HO-1 expression was increased in the NGR1-treated cells compared with the control cells (*p* = 0.029). In the cisplatin-treated cells, the HO-1 level was higher in the cells treated with NGR1 than in the cells not treated with NGR1 (*p* = 0.002). The analysis of HO-1 gene expression revealed that HO-1 expression was upregulated in the NGR1-treated cells compared with the control cells (*p* = 0.02). Among the cisplatin-treated cells, cells treated with NGR1 presented a higher HO-1 mRNA level than cells incubated without NGR1 (*p* = 0.049). These results indicated that NGR1 can induce HO-1 expression in auditory cells. ZNPPIX, an HO-1 inhibitor, was administered to cisplatin-treated cells to investigate the role of HO-1 in the protective effect of NGR1 on cisplatin-induced cytotoxicity (Figure 5C). No significant change in the viability of HEI-OC1 cells was observed following ZNPPIX treatment. The viability of the cells in the group treated with NGR1, cisplatin and ZNPPIX was significantly lower than that of the cells in the groups treated with NGR1 and cisplatin (*p* < 0.001). A significant difference in cell viability was not observed between the groups treated with NGR1, cisplatin and ZNPPIX and the group treated with cisplatin (*p* = 0.749). These results show that the protective effect of NGR1 on cisplatin-induced cytotoxicity can be completely inhibited by ZNPPIX. These findings suggest that NGR1 attenuates cisplatin-induced cytotoxicity in auditory cells by activating HO-1 to reduce oxidative stress.

### 2.3. NGR1 Protects Cochlear Hair Cells from Cisplatin-Induced Ototoxicity

Cochlear explants were pretreated with NGR1 before the administration of cisplatin to evaluate the protective effects of NGR1 on cochlear hair cells damaged by cisplatin (Figure 6). In the group treated with cisplatin alone, a prominent loss of OHCs occurred. Moreover, severe damage to the structure of the stereociliary bundles of inner hair cells and OHCs, including a loss of bundles and prominent disruption, was detected. Compared with the group treated with cisplatin alone, less OHC loss (*p* = 0.009) and better preservation of the stereociliary bundles of hair cells were observed in the group treated with cisplatin and NGR1. These results show that NGR1 protects cochlear hair cells from cisplatin-induced ototoxicity. In addition to oxidative stress playing an important role in the mechanism of cisplatin-induced ototoxicity, iNOS expression triggered by cisplatin leads to cochlear inflammation and RNS production [2]. The ROS levels detected by 4-HNE and the iNOS staining intensity in the organ of Corti were subsequently compared among the three groups (Figure 7). The intensity of 4-HNE staining in inner hair cells and OHCs was higher in the group treated with cisplatin alone than in the group treated with cisplatin and NGR1. These findings reveal that NGR1 can reduce cisplatin-induced ROS generation in cochlear hair cells. Compared with the group treated with cisplatin alone, weaker staining for iNOS was detected in the group treated with cisplatin and NGR1. These findings reveal that NGR1 can inhibit iNOS activation in cisplatin-exposed cochlear hair cells. Taken together, these findings confirm that NGR1 protects cochlear hair cells and attenuates cisplatin-induced ototoxicity by inhibiting oxidative stress and iNOS expression.

## 3. Discussion

Chemotherapy-induced hearing loss and tinnitus lead to poor quality of life and poor daily performance in patients [25,26]. Currently, preventing ototoxicity from cisplatin treatment is still challenging. This study demonstrated that NGR1 has a protective effect on cisplatin-induced cochlear toxicity. The mechanisms of cisplatin-induced ototoxicity involve the pathways of apoptosis, autophagy, mitophagy, the inflammatory response and oxidative stress [2,4,27]. Damaged DNA and mitochondria trigger intrinsic and extrinsic apoptosis pathways, including the activation of caspase-3, -7 and -9 [28,29]. The formation of ROS elicited by damaged subcellular structures promotes apoptosis. Apoptosis in cochlear hair cells leads to the irreversible loss of these cells due to the lack of capacity for hair cell regeneration in adult mammalian inner ears [30]. Autophagy and mitophagy play protective roles in various cochlear injuries. The activation of autophagy and mitophagy attenuates cisplatin-induced OHC damage and hearing loss [27,31,32]. The inflammatory response, including the production of proinflammatory cytokines, translocation of nuclear factor kappa B, and iNOS expression in the cochlea, is elicited by cisplatin [2]. Increased inflammation contributes to ROS generation [33]. Cisplatin-induced ototoxicity is strongly associated with the generation of excessive ROS in the cochlea, which cannot be removed by the endogenous antioxidant defense system [2]. The loss of hearing and OHCs are attributed mainly to increased oxidative stress in the cochlea [33]. This study revealed that NGR1 has antioxidant activity and scavenges ROS in cisplatin-treated HEI-OC1 auditory cells and the cochlea. In the model of cisplatin-induced cytotoxicity, the apoptosis of auditory cells and cochlear OHC loss were ameliorated by NGR1 treatment. Moreover, NGR1 treatment led to the preservation of the mitochondrial mass in cisplatin-treated auditory cells. These findings suggest that NGR1 protects the cochlea from cisplatin-induced mitochondrial damage.

HO-1 expression can be upregulated by various stress stimuli and plays a protective role [8,34]. The induction of HO-1 expression exerts antioxidant, anti-inflammatory and antiapoptotic effects on pathological conditions [8]. Increased HO-1 activity promotes the degradation of heme, a potentially toxic substance, and generates bilirubin and carbon monoxide, which possess antioxidant activity [8]. HO-1 is a pivotal enzyme with antioxidant activity that ameliorates cisplatin-induced oxidative stress in the cochlea [23,24]. The induction of HO-1 expression also attenuates iNOS expression in an animal model of glomerulonephritis [35]. Previous studies have shown that agents that enhance the Nrf2/HO-1 signaling pathway can protect against cisplatin-induced and oxygen-glucose deprivation (OGD)-induced ototoxicity in HEI-OC1 cells and animal models [9,23,24,36]. The pharmacological induction of HO-1 expression suppresses cisplatin-induced apoptosis and oxidative stress in HEI-OC1 cells. Compared with control cells, HO-1-overexpressing cells are more resistant to cisplatin. Furthermore, the induction of HO-1 expression protects against the cisplatin-induced loss of hair cells in the cochlear explants of rats [9]. Kim et al. suggested that HO-1 plays a significant protective role in various types of oxidative damage to the cochlea [9]. Ginkgolide B and polydatin protect against cisplatin-induced hearing loss by promoting Nrf2 translocation and the expression of HO-1, a downstream gene of Nrf2 [23,24]. Macrophage migration inhibitory factor attenuates OGD-induced oxidative stress and damage in HEI-OC1 cells through the Nrf2/HO-1 pathway [36]. NGR1 can facilitate the nuclear translocation of Nrf2 to subsequently upregulate HO-1. HO-1 upregulation plays an important role in the mechanism by which NGR1 protects against ischemia-reperfusion injury in several organs [37]. NGR1 promotes HO-1 expression to prevent diabetic nephropathy in mice [20]. In this study, HO-1 activity in HEI-OC1 auditory cells was increased by NGR1. NGR1 exerts a protective effect on cisplatin-induced damage in cells by activating HO-1 and its antioxidant activity. The suppression of HO-1 activity by ZNPPIX markedly abolished the protective effect of NGR1 on cisplatin-treated cells. In cochlear explants, NGR1 protects cochlear hair cells and attenuates cisplatin-induced ototoxicity by inhibiting ROS generation and iNOS expression. Taken together, these results suggest that NGR1 treatment protects the cochlea from cisplatin-induced damage via the activation of HO-1 and a reduction in oxidative stress. This study is the first to document the protective effect of NGR1 on cisplatin-induced ototoxicity through HO-1 activation.

NGR1 exhibits anticancer activity and can also enhance the anticancer effects of chemotherapeutic agents [18,38]. *Panax notoginseng* enhances the actions of three chemotherapeutic agents, namely, cisplatin, 5-fluorouracil and irinotecan [39,40]. These findings suggest that *Panax notoginseng* can be utilized as a chemoadjuvant for potentiating the tumoricidal effects of chemotherapeutic agents and for reducing the toxicity of these agents [39]. NGR1 enhances the cytotoxicity of cisplatin in HeLa cells by promoting gap junction formation [40]. Our study showed the protective effect of NGR1 on cisplatin-induced ototoxicity. Therefore, NGR1 could be combined with cisplatin-based chemotherapy to potentiate the therapeutic effect of cisplatin and prevent ototoxicity. Further studies will be conducted to elucidate the role of NGR1 in cisplatin-based chemotherapy.

## 4. Materials and Methods

### 4.1. Cell Culture

HEI-OC1 cells, a mouse auditory cell line, were offered by Dr. Federico Kalinec (House Ear Institute, Los Angeles, CA, USA) and were maintained in Dulbecco’s modified Eagle’s medium (DMEM, Gibco, Thermo Fisher Scientific, Waltham, MA, USA) supplemented with 10% fetal bovine serum (FBS, Thermo Fisher Scientific) [41]. HEI-OC1 cells were cultured at 33 °C with 10% CO_2_. The cells were seeded on a 24-well plate for 3 × 10^4^ cells/well, incubated overnight for attachment, and treated with various concentrations of NGR1 (Sigma-Aldrich, Merck KGaA, Darmstadt, Germany) for 24 or 48 h to evaluate the toxic effects of NGR1 on the cells. Treatment with 20 μM cisplatin (Sigma-Aldrich, Merck KGaA, Darmstadt, Germany) was performed for 24 h to induce cisplatin-induced cytotoxicity. In the NGR1 treatment groups, the cells were pretreated with 40 μM NGR1 for 24 h and then cotreated with cisplatin and NGR1 for another 24 h. HO-1 expression was inhibited by treating the cells with 10 μM zinc protoporphyrin IX (ZNPPIX, Cayman Chemical, Ann Arbor, MI, USA) for 48 h. The cells were collected for analysis after 24 h of cisplatin treatment.

### 4.2. Cochlear Explant Culture

The Institutional Animal Care and Use Committee of the National Defense Medical Center, Taipei, Taiwan, approved the experimental protocols (IACUC-22-067). The animal care protocols complied with all institutional guidelines and regulations. The CBA/CaJ mouse strain was used. A total of 75 neonatal mice were used in this study. The inner ears of neonatal mice were harvested at postnatal day 3 according to the procedures described in our previous study [42]. After euthanasia, the cochlea was carefully removed from the temporal bone. The organ of Corti was precisely separated from the spiral lamina and spiral ligament and then placed in a glass-bottomed dish (Ibidi, Grafelfing, Germany) coated with Cell-Tak (#354240, Corning, Thermo Fisher Scientific, Waltham, MA, USA) for attachment. The explants were incubated (37 °C and 5% CO_2_) for 10–16 h, after which 200 µL of the final culture media (97% DMEM, 1% FBS, and 1% N_2_ supplement [Thermo Fisher Scientific, Waltham, MA, USA] and 1% ampicillin) was added to the explants. The explants were then incubated with 30 μM cisplatin for 24 h. In the NGR1 treatment groups, the explants were pretreated with 40 μM NGR1 for 24 h and then cotreated with cisplatin and NGR1 for another 24 h.

### 4.3. Evaluation of Cell Viability

After cells were treated, the cell proliferation reagent WST-1 (Merck KGaA, Darmstadt, Germany) was added to the suspended cells in each well following the manufacturer’s protocol and incubated for 4 h to assess cell viability. The reaction was catalyzed by a mitochondrial reductase in the active cells, and the amount of formazan dye was quantified. The absorbance was measured at 450 nm using an ELISA microplate reader (Synergy H4 Hybrid Reader, Agilent Technologies, Santa Clara, CA, USA) to calculate the optical density (OD) values (A450–A655 nm).

### 4.4. Apoptosis Detection

A TUNEL assay (Merck KGaA, Darmstadt, Germany), immunofluorescence staining of cleaved caspase-3, and Western blot analysis of cleaved caspase-3 were performed to evaluate apoptosis in the cells. In a TUNEL assay, the cells were fixed in a 4% paraformaldehyde solution for 30 min and then incubated with the permeabilization solution for 2 min. After the cells were washed, they were incubated with the TUNEL reaction mixture at 37 °C for 1 h. In the immunofluorescence staining of cleaved caspase-3, the cells were incubated with anti-cleaved caspase-3 antibody (1:500, Cell Signaling, Danvers, MA, USA) in an antibody dilution buffer (Dako, Agilent Technologies, Santa Clara, CA, USA), incubated in a humidified chamber for 1 h and, stained with a secondary antibody (donkey anti-rabbit Alexa Fluor 488, 1:500, Molecular Probes, Thermo Fisher Scientific, Waltham, MA, USA) for an additional 1 h. The cells were then mounted in DAPI Fluoromount-G^®^ mounting medium (SouthernBiotech, Birmingham, AL, USA). The cell images were obtained using a confocal laser scanning microscope (Zeiss LSM 880, Carl Zeiss, Jena, Germany).

### 4.5. Mitochondrial Labeling with MitoTracker Dye

The mitochondrial mass was detected using MitoTracker Green FM (Thermo Fisher Scientific, Waltham, MA, USA). A dilution of the stock solution was applied to the normal growth medium, and the cells were incubated for 30 min. After washing, the cells were incubated with Hoechst 33342 (Thermo Fisher Scientific, Waltham, MA, USA) for 5 min. After the staining was complete, the medium was replaced with fresh medium, and the live cells were imaged using a confocal laser scanning microscope (Zeiss LSM 880, Carl Zeiss, Jena, Germany).

### 4.6. ROS Detection

Cellular ROS levels were measured with the fluorescent dye DCFH-DA (D399; Thermo Fisher Scientific). Briefly, HEI-OC1 cells were seeded in 24-well plates and treated with the indicated reagents. After washes with phosphate-buffered saline (PBS), a medium containing 20 μM DCFH-DA was added to each well and incubated at 33 °C for 30 min. The ROS levels were measured using a microplate ELISA reader (Synergy H4 Hybrid Reader, Agilent Technologies, Santa Clara, CA, USA) at an emission wavelength of 528 nm and an excitation wavelength of 485 nm. The relative ROS levels are presented as the change in fluorescence of the experimental groups compared with that of the control groups. Furthermore, 4-hydroxynonenal (4-HNE) immunofluorescence staining was utilized to detect ROS levels. The cells were incubated with polyclonal primary antibodies against 4-HNE (1:100; Abcam, Cambridge, UK) in BlockPRO^TM^ blocking buffer (Energenesis Biomedical Co., Ltd., Taipei, Taiwan) in a humidified chamber for 1 h and then stained with a secondary antibody (donkey anti-rabbit conjugated to Alexa Fluor 488, 1:500; Molecular Probes; Thermo Fisher Scientific, Waltham, MA, USA) for an additional 1 h. Then, the cells were mounted in DAPI Fluoromount-G^®^ mounting medium. Images were captured with a confocal laser scanning microscope (Zeiss LSM 880, Carl Zeiss, Jena, Germany).

### 4.7. Quantitative PCR

Total RNA was extracted from HEI-OC1 cells using a high-purity RNA isolation kit (Merck KGaA, Darmstadt, Germany) according to the manufacturer’s protocol. The RNA was converted to cDNA using a QuantiNova Reverse Transcription Kit (QIAGEN, Hilden, Germany). The expression of the HO-1 (probe ID: Mm00516005_m1) and glyceraldehyde-3-phosphate dehydrogenase (GAPDH; probe ID: Mm99999915_g1) genes was measured with TaqMan gene expression assays (Thermo Fisher Scientific, Waltham, MA, USA) using a QuantiNova Probe RT-PCR Kit (QIAGEN, Hilden, Germany) and a QuantStudio 5 Real-Time PCR system (Thermo Fisher Scientific, Waltham, MA, USA). The qPCR data are presented as the gene expression level relative to the level in the controls after normalization to the expression of GAPDH.

### 4.8. Western Blotting

Aliquots of cell lysate were separated on mPAGE^TM^ 4–12% Bis-Tris polyacrylamide gels (Merck KGaA, Darmstadt, Germany), transferred to polyvinylidene difluoride membranes (Millipore, Billerica, MA, USA), blocked with BlockPRO^TM^ blocking buffer (Energenesis Biomedical Co., Ltd., Taipei, Taiwan), probed with anti-cleaved Caspase-3 (Cell Signaling Technology, Danvers, MA, USA), anti-HO-1 (Cell Signaling Technology, Danvers, MA, USA), and anti-actin (Merck KGaA, Darmstadt, Germany) antibodies at 4 °C overnight, washed with PBST, and incubated with horseradish peroxidase-conjugated secondary anti-rabbit or anti-mouse antibodies (1:10,000; Cytiva, Marlborough, MA, USA) diluted with BlockPRO^TM^ blocking buffer (Energenesis Biomedical Co., Ltd., Taipei, Taiwan) for 1 h at room temperature (RT). Immunoblot images were acquired with a UVP ChemStudio Imager (Analytik Jena Co., Upland, CA, USA) using Immobilon^®^ Western Chemiluminescent HRP Substrate (ECL; Merck KGaA, Darmstadt, Germany).

### 4.9. Immunofluorescence Staining of Cochlear Explants

The cochlear explants were fixed with 4% paraformaldehyde for 30 min at RT and then incubated with the permeabilization solution for 30 min. After the explants were washed, they were blocked with BlockPRO^TM^ blocking buffer (Energenesis Biomedical Co., Ltd., Taipei, Taiwan). The cochlear explants were incubated with anti-myosin VIIa (Santa Cruz Biotechnology, Dallas, TX, USA), anti-4-HNE (Abcam, Cambridge, UK) and anti-iNOS (Novus Biologicals, LLC, Centennial, CO, USA) antibodies for 2 h at RT. After rinses with PBST, the samples were incubated with Alexa Fluor™ 647-conjugated Phalloidin (1:500; Thermo Fisher Scientific, Waltham, MA, USA) for 30 min. The samples were incubated with DAPI (1:1000; Thermo Fisher Scientific) for 15 min. Images were obtained using an LSM 880 Zeiss confocal microscope (Carl Zeiss, Jena, Germany).

### 4.10. Statistical Analysis

The Shapiro–Wilk test was used to confirm the obtained data were normally distributed. The data were analyzed statistically using Student’s t-test for comparisons between the two groups. Multiple groups were compared using one-way ANOVA followed by Scheffe’s multiple comparisons test. The results are presented as the means ± standard errors of the means (SEMs). Differences were considered significant at *p* < 0.05.

## 5. Conclusions

We confirmed that NGR1 has a protective effect on cisplatin-induced damage to auditory cells both in vitro and ex vivo by increasing HO-1 expression. In cochlear explants, the administration of NGR1 ameliorated oxidative stress in the cochlea and led to less OHC loss and better preservation of the stereociliary bundles of hair cells. These findings suggest that NGR1 could have clinical applications in the prevention of cisplatin-induced ototoxicity.

## Figures and Tables

**Figure 1 ijms-25-11444-f001:**
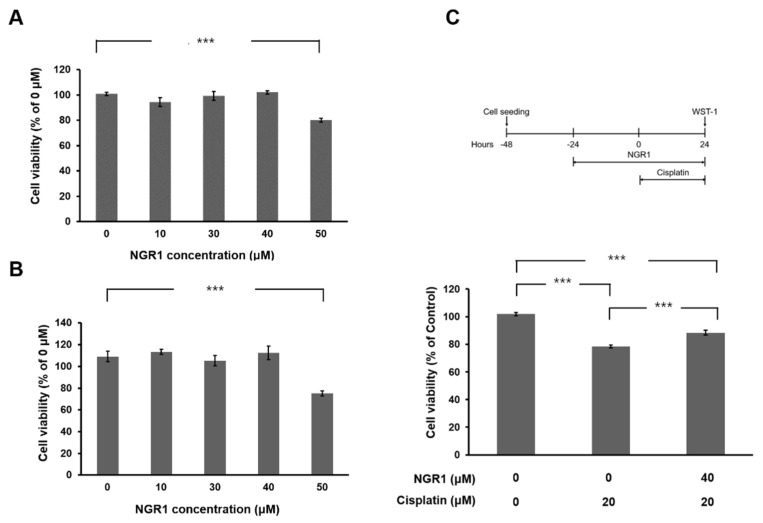
Cisplatin-induced cytotoxicity in HEI-OC1 auditory cells can be attenuated by treatment with notoginsenoside R1 (NGR1). (**A**) Viability of HEI-OC1 cells after 24 h of treatment with various concentrations of NGR1. *n* = 12 for each group. (**B**) Viability of HEI-OC1 cells after 48 h of treatment with various concentrations of NGR1. *n* = 12 for each group. (**C**) Viability of HEI-OC1 cells treated with NGR1 and cisplatin. The experimental workflow is shown at the top. The cells were treated with 20 μM cisplatin alone for 24 h or pretreated with NGR1 for 24 h, followed by 24 h of cotreatment with NGR1 and cisplatin. *n* = 30 for each group. *** *p* < 0.005.

**Figure 2 ijms-25-11444-f002:**
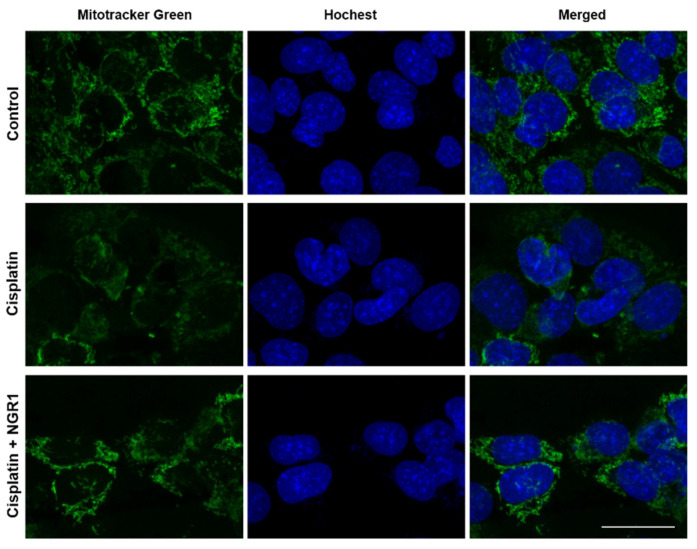
Representative confocal images of MitoTracker Green FM mitochondrial staining in HEI-OC1 cells after cisplatin and NGR1 treatments. The cells were treated with 20 μM cisplatin alone for 24 h or pretreated with NGR1 for 24 h, followed by 24 h of cotreatment with NGRI and cisplatin. Green indicates mitochondria in the cells. Blue indicates Hochest-stained nuclei. *n* = 4 for each group. Scale bar: 25 μm.

**Figure 3 ijms-25-11444-f003:**
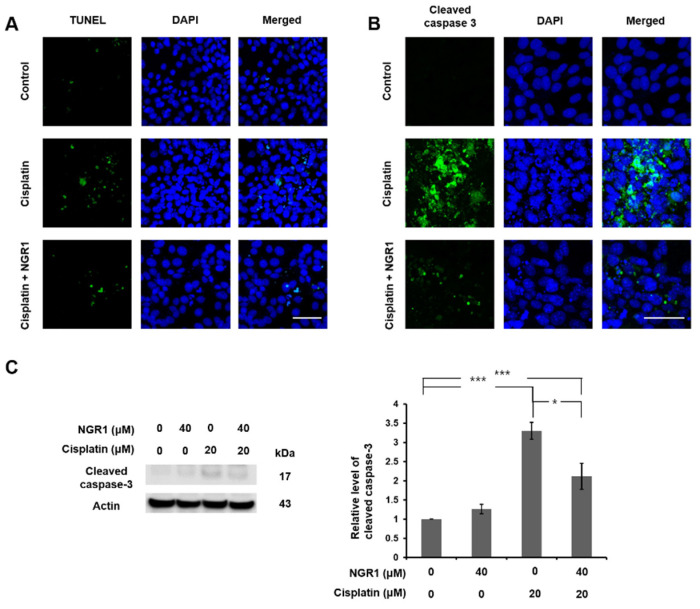
NGR1 ameliorates the cisplatin-induced apoptosis of HEI-OC1 cells. The cells were treated with 20 μM cisplatin alone for 24 h or pretreated with NGR1 for 24 h, followed by 24 h of cotreatment with NGRI and cisplatin. (**A**) Representative image of the TUNEL assay in HEI-OC1 cells. Green highlights TUNEL-positive cells. Blue indicates DAPI-stained nuclei. *n* = 6 for each group. Scale bar: 50 μm. (**B**) Representative image of immunostaining for cleaved caspase-3 in HEI-OC1 cells. *n* = 6 for each group. Scale bar: 50 μm. (**C**) Western blot analysis of cleaved caspase-3 levels in HEI-OC1 cells. *n* = 6 for each group. * *p* < 0.05, and *** *p* < 0.005.

**Figure 4 ijms-25-11444-f004:**
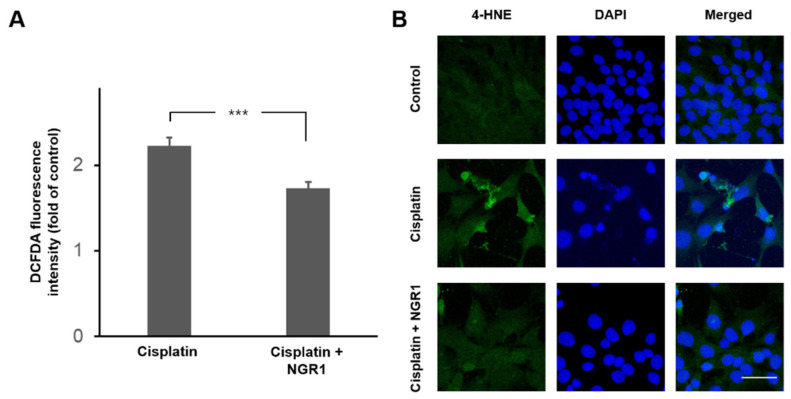
Cisplatin-induced increases in ROS levels in HEI-OC1 cells were reduced by NGR1 treatment. (**A**) ROS generation was measured using the fluorescent probe DCFDA. The results are presented as the means ± SEMs. *n* = 12 for each group. *** *p* < 0.005. (**B**) Representative image of immunostaining for 4-HNE (green) in HEI-OC1 cells. Blue indicates Hochest-stained nuclei. *n* = 4 for each group. Scale bar: 50 μm.

**Figure 5 ijms-25-11444-f005:**
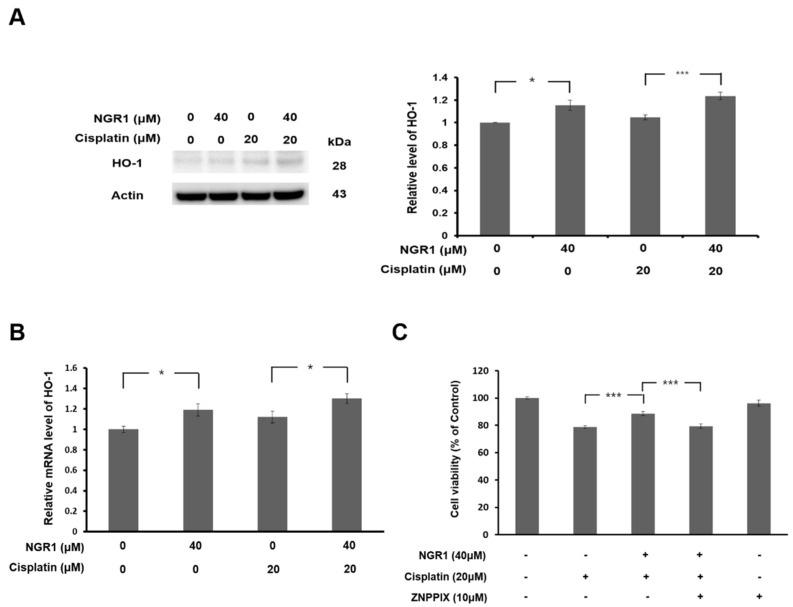
NGR1 attenuated cisplatin-induced cytotoxicity in HEI-OC1 cells by inducing HO-1 expression. (**A**) Western blot analysis of HO-1 levels in HEI-OC1 cells. The cells were treated with 20 μM cisplatin for 24 h or NGR1 for 48 h or pretreated with NGR1 for 24 h followed by 24 h of cotreatment with NGRI and cisplatin. *n* = 5 for each group. (**B**) qPCR analysis of HO-1 mRNA expression in HEI-OC1 cells. The cells were treated with cisplatin for 4 h, NGR1 for 24 h or pretreated with NGR1 for 24 h followed by 4 h of cotreatment with NGRI and cisplatin. *n* = 6 for each group. (**C**) ZNPPIX, a HO-1 inhibitor, attenuated the protective effect of NGR1 on cisplatin-induced cytotoxicity in HEI-OC1 cells. *n* = 22 for each group. * *p* < 0.05 and *** *p* < 0.005.

**Figure 6 ijms-25-11444-f006:**
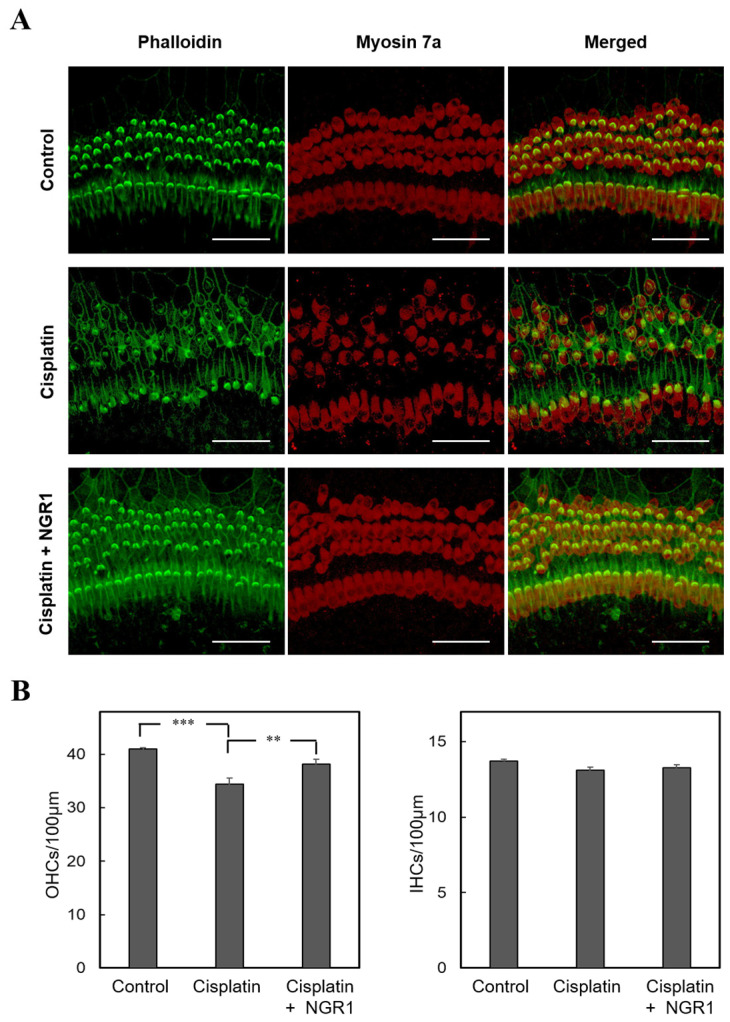
NGR1 protects cochlear hair cells from cisplatin-induced ototoxicity. (**A**) Representative images of organ of Corti explants. The explants were treated with 30 μM cisplatin alone for 24 h or pretreated with 40 μM NGR1 for 24 h, followed by 24 h of cotreatment with NGRI and cisplatin. Phalloidin, stereociliary bundles (green); myosin 7a, cell bodies (red). Scale bar: 50 μm. (**B**) Comparison of number of cochlear hair cells from the different groups. *n* = 18 for each group. ** *p* < 0.01 and *** *p* < 0.005.

**Figure 7 ijms-25-11444-f007:**
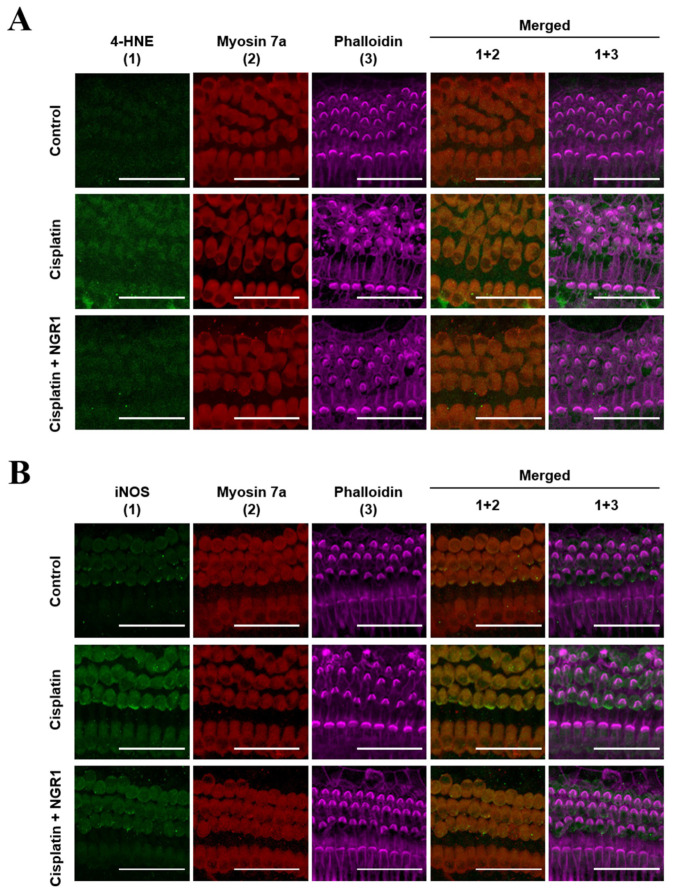
(**A**) Representative images of immunostaining for 4-HNE in the organ of Corti. *n* = 4 for each group. (**B**) Representative images of iNOS immunostaining in the organ of Corti. *n* = 3 for each group. The explants were treated with 30 μM cisplatin for 24 h and pretreated with 40 μM NGR1 for 24 h, followed by 24 h of cotreatment with NGRI and cisplatin. Phalloidin, stereociliary bundles (violet); myosin 7a, cell bodies (red); DAPI, 4-HNE or iNOS (green). Scale bar: 50 μm.

## Data Availability

The data presented in this study are available upon reasonable request from the corresponding author.

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
