# Peer review of "Notoginsenoside R1 Attenuates Cisplatin-Induced Ototoxicity by Inducing Heme Oxygenase-1 Expression and Suppressing Oxidative Stress"

_ijms, 2024, doi:10.3390/ijms252111444_

Round 1

Reviewer 1 Report

Comments and Suggestions for Authors

Cisplatinum induced ototoxicity remains still a major adverse effect in oncology patients treated with this live-saving medication. Every carefully performed study helping to better understand the complex mechanisms involved and to alleviate this iatrogenic burden is welcome.

In the present study Lin and co-workers identify Notoginsenoside R1 (NGR1), the main bioactive compound of Panax notoginseng saponins, is partially protective against cisplatimum induced ototoxicity. In  in vitroand ex vivoexperiments, using the mouse auditory HEI-OC1 cell line and neonatal murine cochlear explants from the CBA/CaJ mouse strainthe authors convincingly show that NGR1 exerts a protective effect through reduction of reactive oxygen species (ROS) via promotion of heme oxygenase-1 (HO-1) expression/activation. Materials & Methods are clearly described, the results well documented by q-PCR, Western blotting and Immunohistochemistry. The Figures are well done and of good photographic quality (immunohistochemistry). The Discussion includes the relevant literature and is to the point.

As to my appreciation, the manuscript will benefit from my minor criticisms and suggestions.

Minor suggestions/criticisms.

1- The manuscript might get more concise and reader-friendly when Figure 3B merged, and especially Figure 6A merged, show an inset done with 63x oil immersion

2- Figures 2, 4 and 7 might be in supplementary material.

Author Response

Point-by-point responses to reviewer comments:

  1. Cisplatinum induced ototoxicity remains still a major adverse effect in oncology patients treated with this live-saving medication. Every carefully performed study helping to better understand the complex mechanisms involved and to alleviate this iatrogenic burden is welcome. 

In the present study Lin and co-workers identify Notoginsenoside R1 (NGR1), the main bioactive compound of Panax notoginseng saponins, is partially protective against cisplatimum induced ototoxicity. In  in vitroand ex vivoexperiments, using the mouse auditory HEI-OC1 cell line and neonatal murine cochlear explants from the CBA/CaJ mouse strain, the authors convincingly show that NGR1 exerts a protective effect through reduction of reactive oxygen species (ROS) via promotion of heme oxygenase-1 (HO-1) expression/activation. Materials & Methods are clearly described, the results well documented by q-PCR, Western blotting and Immunohistochemistry. The Figures are well done and of good photographic quality (immunohistochemistry). The Discussion includes the relevant literature and is to the point.

As to my appreciation, the manuscript will benefit from my minor criticisms and suggestions.

Minor suggestions/criticisms.

1- The manuscript might get more concise and reader-friendly when Figure 3B merged, and especially Figure 6A merged, show an inset done with 63x oil immersion

Reply: Thanks for your comment and good suggestion. In this study, all immunofluorescence images were obtained using 40x objective in an LSM 880 Zeiss confocal microscope. There is no image obtained with 63x oil immersion objective in this study. Therefore, an inset done with 63x oil immersion cannot be added in the figure. We will use 63x oil immersion and confocal microscope to collect immunofluorescence images in the future.

  1. Figures 2, 4 and 7 might be in supplementary material.

Reply: Thanks for your suggestion. It may be more convenient to read the article when figures 2, 4 and 7 still remain in the text. Therefore, we would like to preserve these figures in the text.

Reviewer 2 Report

Comments and Suggestions for Authors

The objective of this study was to evaluate the effect of Notoginsenoside R1 (NGR1) on attenuating the cisplatin-induced ototoxicity and associated mechanism in HEI-OC1 mouse auditory cells. The results demonstrated that NRG1 treated reduced the cytotoxic effect of cisplatin and cisplatin-induced intracellular ROS generation and promoted the heme oxygenase-1 (HO-1) expression in HEI-OC1 cells. It was concluded that NGR1 exhibited inhibitory effect on

cisplatin-induced ototoxicity by elevating HO-1 expression and suppressing oxidative stress.

Major comments:

1. Notoginsenoside R1 has a molecular weight of 933 Da. The NRG1 concentration used in the in vitro study was 40 mcro-M, i.e., 37.3 mcg/mL or 37.3 mcg/g tissue. I wonder if the concentration of NRG1 used in the in vitro assessment would be biologically relevant.

2. Figure 1. What was the rationale of using 40 micro-M of NGR1 in the experiment? The authors should include NGR1 at lower concentrations, e.g., 5, 10 and 20 micro-M to determine if the protective effect of NGR1 against cisplatin-induced oxotoxicity is concentration-dependent.

3. In all of the experiments, HEI-OC1 cell or the Corti explants was pre-treated with NGR1. Would pretreatment with NGR1 compromise the anti-cancer effect of cisplatin? The authors should consider including a tumor cell line in the study for comparison purposes.

Author Response

Point-by-point responses to reviewer comments:

  1. The objective of this study was to evaluate the effect of Notoginsenoside R1 (NGR1) on attenuating the cisplatin-induced ototoxicity and associated mechanism in HEI-OC1 mouse auditory cells. The results demonstrated that NRG1 treated reduced the cytotoxic effect of cisplatin and cisplatin-induced intracellular ROS generation and promoted the heme oxygenase-1 (HO-1) expression in HEI-OC1 cells. It was concluded that NGR1 exhibited inhibitory effect on cisplatin-induced ototoxicity by elevating HO-1 expression and suppressing oxidative stress.

Major comments:

  1. Notoginsenoside R1 has a molecular weight of 933 Da. The NRG1 concentration used in the in vitro study was 40 mcro-M, i.e., 37.3 mcg/mL or 37.3 mcg/g tissue. I wonder if the concentration of NRG1 used in the in vitro assessment would be biologically relevant.

Reply: Thanks for your comment. The review article by Liu et al. (Reference 18 in our article: Liu, H.; Yang, J.; Yang, W.; et al. Focus on notoginsenoside R1 in metabolism and prevention against human diseases. Drug Des. Devel. Ther. 2020, 14, 551-565) reports that Notoginsenoside R1 (NGR1) with various tested concentrations (5~50 μM), including 40 μM has protective effect on different diseases in several in vitro and in vivo studies. Furthermore, the article mentions that protection of NGR1 against diseases in the animal studies in which 20~30 mg/kg (20~30 mcg/g) NGR1 is administered. This present study showed 40 μM NRG1 can protect against cisplatin-induced ototoxicity. Therefore, we think 40 μM NRG1 used in the in vitro assessment is biologically relevant.

  1. Figure 1. What was the rationale of using 40 micro-M of NGR1 in the experiment? The authors should include NGR1 at lower concentrations, e.g., 5, 10 and 20 micro-M to determine if the protective effect of NGR1 against cisplatin-induced oxotoxicity is concentration-dependent.

Reply: Thanks for your comment. First, we tested 30 μM NGR1 for cell viability experiment of cisplatin-induced cytotoxicity in HEI-OC1 cells. The WST-1 result showed no significant difference in cell viability between the group with cisplatin alone and the group with cisplatin and 30 μM NGR1 (78.7% ± 1.48% vs. 79.2% ± 1.62%, p = 0.8217). Therefore, we did not test NGR1 with concentration below 30 μM. Subsequently, we investigated the effect of 40 μM NGR1 on cisplatin-induced cytotoxicity. A significant difference was observed between cisplatin alone and the group with cisplatin and 40 μM NGR1 (78.4% ± 1.12% vs. 88.4% ± 1.65%, p < 0.001). The WST-1 result of the group with cisplatin and the group with cisplatin and 30 μM NGR1 was added in Supplementary Figure S1.

In page 3 line 101 - 106 of the revised manuscript (changes are highlighted in red):

First, the effect of 30 μM NGR1 on cell viability in cisplatin-treated HEI-OC1 cells was investigated (Supplementary Figure S1). There was no significant difference in cell viability between the group with cisplatin alone and the group with cisplatin and 30 μM NGR1 (p = 0.8217). Therefore, the concentration of NGR1 used in subsequent experiments was 40 μM. The protective effect of 40 μM NGR1 on cisplatin-induced cytotoxicity was investigated (Fig. 1C).

Supplementary Figure S1:

Figure S1. Viability of HEI-OC1 cells treated with 30 μM NGR1 and cisplatin. The cells were treated with 20 μM cisplatin alone for 24 h or pretreated with NGR1 for 24 h, followed by 24 h of cotreatment with NGR1 and cisplatin. n = 12 for each group.

  1. In all of the experiments, HEI-OC1 cell or the Corti explants was pre-treated with NGR1. Would pretreatment with NGR1 compromise the anti-cancer effect of cisplatin? The authors should consider including a tumor cell line in the study for comparison purposes.

Reply: Thanks for your comment and suggestion. The aim of this present study is to investigate the ability of NGR1 to protect against cisplatin-induced ototoxicity. We have demonstrated that NGR1 can protect against cisplatin-induced ototoxicity. In discussion section of the manuscript (page 10 line 282 – page 11 line 291), anti-cancer effect of chemotherapeutic agents enhanced by NGR1 has been descripted. The previous studies show that NGR1 exhibits anticancer activity and can also enhance the anticancer effects of chemotherapeutic agents [reference 18, 38 in our article]. NGR1 enhances the cytotoxicity of cisplatin in HeLa cells by promoting gap junction formation [reference 40 in our article]. According to the findings in previous studies and this present study, NGR1 may potentiate the anti-cancer effect of cisplatin and prevent ototoxicity induced by cisplatin. Therefore, pretreatment with NGR1 might not compromise the anti-cancer effect of cisplatin. My colleague, Dr. Hang-Kang Chen, has established squamous cell cancer of the head and neck-bearing mouse model in his previous study. We will further conduct an animal study to elucidate the role of NGR1 in cisplatin-based chemotherapy, including anti-cancer effect and preventive effect on cisplatin-induced ototoxicity.
